# A VR Prototype for One-Dimensional Movement Visualizations for Robotic Arms

### Bram van Deurzen
UHasselt - Flanders Make
Expertise Center for Digital Media
Diepenbeek, Belgium
bram.vandeurzen@uhasselt.be

### Dries Cardinaels
UHasselt - Flanders Make
Expertise Center for Digital Media
Diepenbeek, Belgium
dries.cardinaels@uhasselt.be

### Gustavo Rovelo Ruiz
UHasselt - Flanders Make
Expertise Center for Digital Media
Diepenbeek, Belgium
gustavo.roveloruiz@uhasselt.be

### Kris Luyten
UHasselt - Flanders Make
Expertise Center for Digital Media
Diepenbeek, Belgium
kris.luyten@uhasselt.be

## ABSTRACT

To enable effective communication between users and autonomous robots, it is crucial to have a shared understanding of goals and actions. This is made possible through an intelligible interface that communicates relevant information. This intelligibility enhances user comprehension, enabling them to anticipate the robot's actions and respond appropriately. However, because robots can perform a wide variety of actions and communication resources are limited, such as the number of available "pixels", visualizations must be carefully designed. To tackle this challenge, we have developed a visual design framework. Leveraging Unity, we developed a Virtual Reality implementation to prototype and evaluate our framework. Within this framework, we introduce two visualization techniques for visualizing the movement of a robotic arm, laying a foundation for subsequent development and user testing.

## CCS CONCEPTS

• **Human-centered computing** → **Visualization theory, concepts and paradigms**; *Displays and imagers*.

## KEYWORDS

Human-Robot Interaction; Intelligibility; Visualizations; Visual Design Framework

ACM Reference Format:
Bram van Deurzen, Dries Cardinaels, Gustavo Rovelo Ruiz, and Kris Luyten. 2024. A VR Prototype for One-Dimensional Movement Visualizations for Robotic Arms. In *Proceedings of ACM Conference (Conference'17)*. ACM, New York, NY, USA, 5 pages. https://doi.org/10.1145/nnnnnnn.nnnnnnn

## 1 INTRODUCTION

Fluent interaction between a user and an autonomous robot requires a clear understanding of each other's intentions, expectations, requirements, and actions [10]. To a large extent, this understanding can be achieved through an *intelligible interface*, which ensures that the robot communicates information to its users effectively. As a result, users can anticipate the robot's actions more accurately, leading to a more predictable and reliable interaction. However, given the diversity of robot actions and often limited resources–in terms of available "pixels"–to communicate with users, carefully designed visualizations are required to inform users without interfering with normal operations. One important consideration is information density because too much information can cause information overload [3, 8]. This can make the user feel overwhelmed and unable to understand the message properly.

Several studies have been conducted on different visualizations of various types of robots and their interactions. Pascher et al. [11] conducted a survey paper on communicating robot motion intent, highlighting some of the work done in this area. Similarly, Yu et al. [18] tested the impact of robot intent visualizations on trust and robot understandability. Collet and MacDonald [6] proposed visualizations to externalize the internal state and robot programming to the user. Sonawani et al. [14] studied projecting the intent of a robotic arm on the collaborative work surface.

The literature provides valuable insights into how to visualize robot intelligibility but is less focused on determining when and how to visualize it. Deciding when and how to visualize this intelligible information is not easy. Nonetheless, it is of utmost importance, especially for robot interaction developers who might not have experience with visual design. Creating usable, intelligible visualizations requires two key components: when and how to show the intelligible visualizations.

This workshop paper focuses on how to create intelligible movement visualizations. We propose a visual design framework that can be used to create intelligible visualizations for human-robot interactions. Based on this visual design framework, we implemented a Virtual Reality (VR) playground to test our intelligible visualizations for a robotic arm. This prototype aims to help us further develop our visual design framework and use it for user studies. Doing our user studies in VR allows us to test a broad range

of interaction scenarios without putting our participants in any dangerous situations.

## 2 VISUAL DESIGN FRAMEWORK

We have developed a visual design framework that consists of three main components that influence the intelligible visualizations. These components are:

- *Information Type*: the type of information that the robot system wants to communicate to the user.
- *Pixel Layout*: the number of available pixels and their layout to create the visualization.
- *Robot Type*: The type of robot being used.

Each of these components is determined by the specific human-robot interaction action that they belong to.

### 2.1 Information Type

The first component of our framework is the information type that the robot needs to externalize to the world. We define three main categories of information types inspired by the intent type defined by Pascher et al. [11].

- *Movement*: All information related to the movement of the robotic system.
- *Interaction*: All information related to interaction between the robot(s) and the human(s).
- *State*: All information related to the robotic system's internal and external state.

The information type that the robot needs to convey to the user influences the intelligible visualizations. For example, movement information can be visualized by indicating the path the robot will take [4, 16]. An example of interaction information can be when input is required, e.g., the robot waiting on the user to open a door for them [17]. An example of an internal state information can be an error state the robot is in [7]. The external state can be an obstacle detected by the robot [6].

Other types of information are possible, but they can be reduced to a combination of our three categories. For example, the perception of a robot system is the combination of the internal and external state.

### 2.2 Pixel Layout

The Pixel Layout represents the visualizations that abstract away the underlying visualization technologies that might change in the future. We define three categories of pixel layout:

- *One Dimensional (1D)*: A single pixel or a single row of pixels. Examples are a brake light, a turn signal, or a progress bar.
- *Two Dimensional (2D)*: A matrix of pixels, i.e., a screen, which can be any size, from a digital number display up to a high-resolution TV screen or a projection.
- *Three Dimensional (3D)*: A display that utilizes Augmented or Mixed Reality technologies to utilize the complete environment.

Since the pixel layout determines how much information can be shown, it influences the visual design. Examples of visualization with a 1D pixel layout include signals in a car or mobile robots [9],

an expressive light to communicate the robot state [1], or directionality in flying robots [15]. Example visualizations with a 2D pixel layout are screen-based visualizations [13] or projections in the human-robot interaction surface [14, 17]. Examples of visualizations with a 3D pixel layout are Augmented Reality (AR) based visualizations of robot path [2, 12] or robot programming using AR [5].

It is important to decide the appropriate type of pixel layout for a specific human-robot action. We theorize that there are three main factors to consider: the criticality of the action, the role of the human during the action, and the robot's proximity to the human. The aim is to maximize the information transfer rate between the robot and the human while minimizing information overload. Depending on the scenario, the most suitable pixel layout should be selected based on the best visualization options available.

### 2.3 Robot Type

The last component is the type of robot used in the human-robot interaction. The abstraction represents a broader set of robot systems while focusing on the key aspects of each robot type. Robot systems can exist out of a combination of these robot types.

- *Mobile Robot*: All variants of robot systems that can move around their environment with direct contact with a surface. From automated guided vehicles (AGV) to self-driving vehicles.
- *Robotic Arm*: Any robot manipulator with at least two degrees of freedom. From a pick-and-place robot to a 7-DOF robotic arm or up to a mechanical human arm.
- *Flying Robot*: All types of robots that can move around their environment without direct contact with a surface, e.g., a drone.

The robot type affects where the visual designs can be displayed on the robot and in the environment. A mobile robot requires an integrated visualization system for mobility, whereas a stationary robotic arm can utilize a fixed visualization setup within its operating environment. The placement of the visualization on the robot is another factor to consider. For instance, on a quadcopter, the weight constraints impact the type of visualization devices that can be used. Similarly, the end effector of a robotic arm has limited space for visualization devices, especially with a higher pixel layout. Large mobile robots, such as AGVs or self-driving vehicles, have more space to put visualization devices of higher degrees of pixel layout. Some already contain these devices, such as turn signals or brake lights.

## 3 ONE DIMENSIONAL MOVEMENT VISUALIZATIONS FOR ROBOTIC ARMS

For this workshop paper, we focus on further developing the 1D pixel layout movement visualizations for robotic arms. We do this by creating VR prototypes of the robot and visualizations in Unity. It allows us to quickly iterate and develop the visualizations themselves and reuse the same environment for future VR user studies.

We first focus on implementing movement for robotic arms instead of mobile or flying robots, as we believe robotic arms are the most interesting use case. Mobile and flying robots can reuse a system very similar to that of car direction indicators. It is a

well-known and broadly used way of representing movement information that can be reused for our design case. For robotic arms, there is no clear preexisting movement visualization.

### 3.1 Designing the Pixel Visualization

In the case of a robotic arm, each joint of the robot influences the movement of the robotic arm. Therefore, we want to be able to visualize information for each joint of the robotic arm independently. Secondly, the robotic arm can move in any direction, so the visualization should be visible from as many directions as possible. Therefore, we mount pixels around the joints of the robot itself.

This results in a robotic arm with a band of pixels added to every joint. These are mounted as close as possible to the rotation point to mirror the movement of the joint. Figure 1 shows a virtual KUKA LBR iiwa[1] robot with the pixel bands added to it.

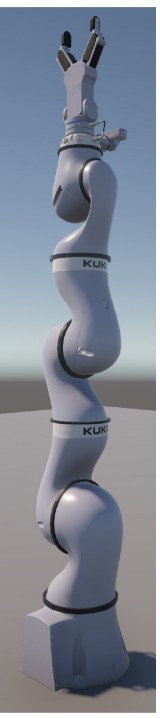

**Figure 1: An example of a virtual robotic arm with the 1D pixel layout visualization added. For each joint, a pixel band is added.**

Each pixel band is subdivided into eight pixels corresponding to the cardinal directions (see Figure 2). It allows us to link the robot's movement in a specific direction with the visualization of the pixel bands. As a robotic arm joint can rotate around a horizontal or a vertical axis, the Unity axis that corresponds with the north direction changes. A horizontal joint rotates around the Y-axis with minus Z as the north direction (Figure 2a). A vertical joint rotates around the Z-axis with Y as the north direction (Figure 2b).

---

[1]Kuka LBR iiwa website

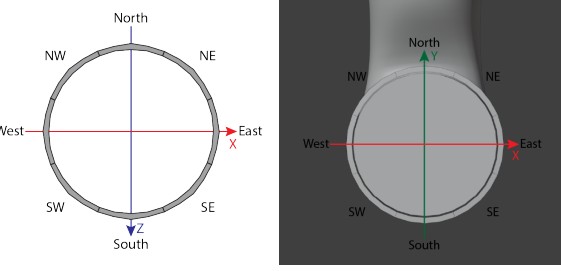

**(a) The horizontal joint.**  **(b) The vertical joint.**

**Figure 2: The subdivision of a pixel band into eight pixels linked to the cardinal directions shown for a horizontal and vertical joint. The axes represent the corresponding Unity axis.**

### 3.2 Movement Information Visualization

To create the intelligible movement visualization, we must know the robot's future movement. Based on this information the visualization can determine where the robot will move to and visualize this movement accordingly. We identified two options to visualize the movement information: angle-based and direction-based.

The angle-based visualization lights up the pixel for each joint corresponding to the angle in which the joint will rotate (see Figure 3). The joints that do not change their rotation during the movement light up the pixel corresponding to their current rotation angle. Each joint visualizes its movement independently from each other.

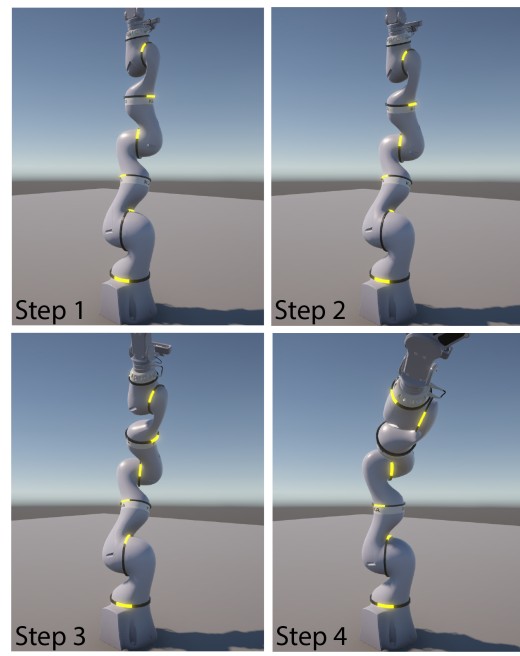

**Figure 3: Storyboard example of the angle-based movement visualization. Four steps of the robot moving towards the user are shown.**

The direction-based visualization lights up the pixel on each joint that corresponds to the direction the robot will move to (see Figure 4). This results in the pixels working in unison to create a vertical direction indicator on the robotic arm corresponding to the robot's movement direction.

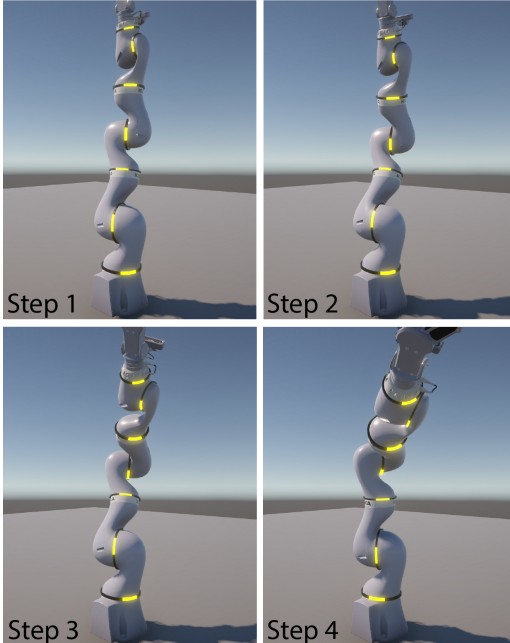

**Figure 4: Storyboard example of the direction-based movement visualization. Four steps of the robot moving towards the user are shown.**

Based on our Unity implementation of the visualizations, we can already see that the angle-based visualization is more prone to obstructions from the robot. On the other hand, direction-based visualization allows us to see the visualization from all directions because when no visible pixels light up, we can deduce that the robot is moving away from the user.

A possible improvement to the angle-based visualization is only to light up the pixel on a joint if it changes its rotation. In this way, the amount of visual information provided to the user is reduced. A light appearing on a joint indirectly signals that the joint will move.

We theorize that the angle-based visualizations are more challenging to interpret as each joint has to be interpreted individually by the user. One change in orientation of a single joint of the robot also moves all joints down the chain of joints in the robot. This can create situations where predicting the robot's movement based on the rotation of a single joint requires additional reasoning by the user. Further development and user testing of this hypothesis is required before we can make concrete statements about this.

## 4 DISCUSSION

In this workshop paper, we explored one-dimensional movement visualizations for robotic arms as part of developing our visual design framework for intelligible human-robot interaction visualizations. We focus on developing a Unity implementation of this visual design framework as a basis for our future research and user testing. The Unity environment enables quick prototyping and VR user testing of our visual design framework.

In a future iteration, we want to add pixel bands to a broader set of robotic arms. We chose the Kuka iiwa robot as a starting point due to its seven degrees of freedom, providing a large number of joints to visualize. Secondly, it also has a specific design focused on minimizing possible pinching points on the robot. This design can cause more visual obstructions by the robot itself. The second robot we want to add to our implementation is the Universal Robots UR10e[2]. It is a robotic arm with six degrees of freedom with a straightforward design, often used in human-robot interaction research.

Our visual design framework aims to support all three types of information, and thus not only movement. We envision reusing the same pixel bands but using different colors to represent state and interaction information. For the end effector itself, we can add extra pixels based on the end effector in use, enabling us to relay information specific to the end effector. For example, indicating the opening or closing of the end effector or when the robot is ready to hand over an object.

To finalize our research, we plan to conduct user studies in VR. It enables us to reuse our existing prototypes with the main benefit of the allowance to test out potentially dangerous scenarios. For example, scenarios where a robotic arm moves in close approximation with the user but where it is important that the user still feels safe. The central hypothesis we want to test is which visualization is the easiest to interpret and creates the highest trust between the user and the robot. To do this, we will compare the angle and direction-based visualization with different robots and in different scenarios.

We show how simple pixels can be mounted on a robotic arm and how they can potentially convey the robot's movement information without the need for a projection or Augmented Reality system.

## ACKNOWLEDGMENTS

This work was funded by the Flemish Government under the "Onderzoeksprogramma Artificiële Intelligentie (AI) Vlaanderen" programme, and by the Special Research Fund (BOF) of Hasselt University, BOF20OWB23.

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
