# OpenReview forum: "A VR Prototype for One-Dimensional Movement Visualizations for Robotic Arms"
_humanrobotinteraction.org/HRI/2024/Workshop/VAM-HRI — VAM-HRI 2024 Oral_

### Official Review · Reviewer_rxyF · 2024-02-20
**Accept**

**Rating:** 8
**Confidence:** 4

**Review:**

The paper explores the development of a virtual reality (VR) prototype designed to visualize the movements of robotic arms through one-dimensional (1D) pixel representations. This innovative approach aims to enhance user comprehension and interaction with robotic systems by providing intuitive and easy-to-understand visual cues within a VR environment.

## Strengths:
- **Innovative Visualization Method:** Introduces a unique concept of using 1D pixel layouts to represent robotic movements, potentially improving user understanding and interaction.
- **VR Integration:** Utilizes VR technology to immerse users in an interactive environment, enhancing the perception of robotic movements.

## Weaknesses:
- **Limited Application Scope:** The focus on 1D visualizations might restrict the complexity of movements that can be effectively represented.
- **User Experience Evaluation:** The paper could benefit from more extensive user testing to evaluate the effectiveness and intuitiveness of the visualizations.

## Recommendations for Improvement:
- **Expand Visualization Techniques:** Incorporate multi-dimensional visualizations to represent more complex robotic movements.
- **Comprehensive User Studies:** Conduct detailed studies to assess user experience, comprehension, and the prototype's overall effectiveness in various scenarios.
- **Enhance Interactivity:** Introduce interactive elements that allow users to manipulate visualizations directly, providing a deeper understanding of robotic movements.

In summary, I think this paper is a great fit for VAM-HRI, and I recommend acceptance.

---

### Official Review · Reviewer_nAXa · 2024-02-24
**Accept**

**Rating:** 8
**Confidence:** 4

**Review:**

The paper presents a visualization framework to communicate the robot's actions and decisions to the human user. The proposed framework addresses the challenge of communicating robot actions effectively, considering limitations such as the number of available "pixels" for visualization, information overload to the user, etc. The idea of this visual design framework consists of 3 components: information type, pixel layout, and robot type. This paper focuses on a specific combination of these components, i.e., 1D pixel layout, and movement type visualization for robotic arm. In the case of the robotic arm, the 1D pixel layout is achieved by attaching a pixel band to each joint. Two methods were identified to visualize the movement of the selected robot arm, i.e. angle-based and direction-based. In the case of angle-based, each joint visualizes its movement independently from the other and lights up corresponding to the angle the joint will rotate. The direction-based visualization lights up the pixel on each joint that corresponds to the direction the robot will move.


Strengths:

1. Innovative approach to robot-human interaction through intelligible visualizations.
2. Comprehensive framework considering different types of robotic systems.

Areas of Improvement:

1. Further user studies are needed to validate the effectiveness of visualizations in real-world scenarios.
2. The paper might benefit from a more detailed discussion on the scalability of the visual design framework to complex movements, or multiple robotic arms operating simultaneously. This would also be the case for other robotic types mentioned.
3. An explanation regarding why the combination of the three choices for the component types for the framework would be beneficial. Would it benefit from additional considerations, such as task type, action type, environment type, and similar?

In summary, I think this paper is a good fit for VAM-HRI, and I recommend acceptance.

---

### Decision · Program_Chairs · 2024-02-26

Accept (Oral)